# Risk Factors for *Pneumocystis jirovecii* Pneumonia in Non-HIV Patients Hospitalized for COVID-19: A Case-Control Study

**DOI:** 10.3390/jof9080838

**Published:** 2023-08-11

**Authors:** Giulio Viceconte, Antonio Riccardo Buonomo, Alessia D’Agostino, Maria Foggia, Antonio Di Fusco, Biagio Pinchera, Riccardo Scotto, Carmine Iacovazzo, Luca Fanasca, Gaetana Messina, Francesco Cacciatore, Paola Salvatore, Ivan Gentile

**Affiliations:** 1Department of Clinical Medicine and Surgery, University of Naples “Federico II”, Via Sergio Pansini n.5, 8031 Naples, Italy; antonioriccardobuonomo@gmail.com (A.R.B.); alessiadagostino53@gmail.com (A.D.); mariafoggia@alice.it (M.F.); difuscoantonio92@gmail.com (A.D.F.); biapin89@virgilio.it (B.P.); ri.scotto@gmail.com (R.S.); ivan.gentile@unina.it (I.G.); 2Department of Neurosciences, Reproductive Sciences and Odontostomatology, University of Naples “Federico II”, Via Sergio Pansini n.5, 8031 Naples, Italy; carmine.iacovazzo@unina.it; 3Department of Molecular Medicine and Medical Biotechnology, University of Naples “Federico II”, 8031 Naples, Italy; luca.fanasca@gmail.com (L.F.); paola.salvatore@unina.it (P.S.); 4Thoracic Surgery Unit, University of Campania “Luigi Vanvitelli”, Via Sergio Pansini n.5, 8031 Naples, Italy; adamessina@virgilio.it; 5Department of Translational Medical Sciences, University of Naples “Federico II”, Via Sergio Pansini n.5, 8031 Naples, Italy; francesco.cacciatore@unina.it

**Keywords:** COVID-19, *Pneumocystis jirovecii*, SARS-CoV-2, immunocompromised, pneumonia

## Abstract

Background: Very few cases of *Pneumocystis jirovecii* pneumonia (PJP) have been reported in COVID-19 so far, and mostly in patients with concomitant HIV infection or in solid-organ transplant recipients. Despite COVID-19 being associated with lymphopenia and the use of steroids, there are no studies specifically aimed at investigating the risk factors for PJP in COVID-19. Methods: A retrospective case-control study was performed. We matched PJP cases with controls with a 1:2 ratio, based on age ± 10 years, solid-organ transplantation (SOT), hematological malignancies, and in the setting of PJP development (ICU vs. non-ICU). A direct immunofluorescence assay on bronchoalveolar lavage fluid was used to diagnose PJP. Results: We enrolled 54 patients. Among 18 cases of PJP, 16 were diagnosed as “proven”. Seven of the eighteen cases were immunocompromised, while the other patients had no previous immunological impairment. Patients with PJP had significantly lower median lymphocyte values (*p* = 0.033), longer COVID-19 duration (*p* = 0.014), a higher dose of steroid received (*p* = 0.026), higher CRP values (*p* = 0.005), and a lower SARS-CoV-2 vaccination rate than the controls (*p* = 0.029). Cumulative steroid dose is the independent risk factor for PJP development (OR = 1.004, 95%CI = 1–1.008, *p* = 0.042). Conclusions: PJP develops in COVID-19 patients regardless of immunosuppressive conditions and the severity of disease, and it is correlated to the corticosteroid dose received.

## 1. Background

Coronavirus Infectious Disease 19 (COVID-19) is the disease caused by Severe Acute Respiratory Syndrome Coronavirus 2 (SARS-CoV-2), characterized by a clinical picture ranging from pauci-symptomatic illness to interstitial pneumonia, leading to respiratory failure and intensive care unit (ICU) admission [1].

Differently from influenza pandemics, respiratory coinfections and superinfections are not very common in COVID-19 patients: according to a 2020 meta-analysis by Lansbury and colleagues, in fact, bacterial coinfections account for 7% of COVID-19 coinfections overall, with only sporadically reported fungal infections [2].

On the other hand, a most recent meta-analysis by Kurra and colleagues, published in 2022, has shown that the pooled prevalence of opportunistic infections in COVID-19 patients is 16%, with the highest prevalence of secondary infections among viruses at 33%; 16% among the bacteria subgroup and 6% among the fungi subgroup [3].

Thus, invasive fungal infections (IFIs) are a non-negligible complication of COVID-19, especially among critically ill patients, and are also associated with a significantly higher mortality: 53% in patients with fungal diseases and 31% in patients without fungal diseases (*p* = 0.0387) [3].

The most common IFIs complicating the course of COVID-19 are invasive candidiasis [4] and pulmonary aspergillosis, with the latter becoming a specific clinical syndrome: COVID-19-associated pulmonary aspergillosis (CAPA), defined by the 2020 ECMM/ISHAM consensus criteria [5]. The cumulative incidence of CAPA in the ICU ranged from 1.0% to 39.1%, with 33.0% of deaths attributed to CAPA, according to a large retrospective cohort of 2020 [6]. Another non-negligible and severe IFI has emerged as complication of COVID-19, especially in India: COVID-19-associated mucormycosis (CAM). CAM is a destructive and potentially life-threatening IFI responsible for 7% of deaths in affected patients. It may also occur 4–6 weeks after COVID-19 onset, and it is highly influenced by the decompensated glycemic status in diabetic patients [7].

Other IFIs, such as *Pneumocystis jirovecii* pneumonia (PJP), are only sporadically reported, almost exclusively in case series and case reports, and mainly in previously immunocompromised hosts (concomitant HIV infection or solid-organ transplant recipients) [8,9]. Unfortunately, no definite consensus about COVID-19-associated PJP, or any definite diagnostic criteria, have been published so far, and the existence of PJP associated with COVID-19 has been questioned by the scientific community. While the diagnostic criteria for IFIs published by the European Organization for Research and Treatment of Cancer and the Mycoses Study Group (EORTC/MSGERC), which are designed for immunocompromised hosts, have also been modified to permit the diagnosis of CAPA in immunocompetent patients, the same has never been done for PJP [10].

In fact, it is difficult to establish a true incidence of PJP among hospitalized COVID-19 patients according to the published studies, since not all the available reports adopt standardized diagnostic criteria for PJP definition and most of them are based on the molecular detection of *Pneumocystis jirovecii* on respiratory samples instead of the direct detection of the fungus with an immunofluorescence assay or direct staining. In fact, the two largest and most recent systematic reviews aimed to describe clinical features and risk factors for PJP in COVID-19, both published between the end of January and the beginning of February 2023, consider a different number of cases: specifically, Amstutz et al. describe 69 cases of PJP in COVID-19 in 29 articles, while Sasani et al. 30 cases [11,12]. Moreover, no study in the literature is available to specifically assess the risk factors for PJP development in COVID-19 using rigorous diagnostic criteria for case definition. Accordingly, we conducted a case-control study to identify the risk factors for PJP in HIV-negative patients hospitalized for COVID-19 at our institution.

## 2. Methods

### 2.1. Study Design

The study was conducted in the “Federico II” University Hospital, located in Naples, Italy, the largest University Hospital of Southern Italy and the referral center for COVID-19 in pregnant women for the Campania Region.

We retrospectively evaluated the clinical data of all the patients admitted for COVID-19 at the Infectious Disease ward and Intensive Care Unit of the “Federico II” University Hospital from 1 November 2021 to 30 September 2022 and who were then diagnosed with PJP during the hospital stay according to the diagnostic criteria reported below. Controls were selected among patients admitted for COVID-19 in the same period who did not receive a diagnosis of PJP. For each case, two controls were matched based on age ± 10 years. All the possible controls were stratified according to the following variables: (i) absence of any previous immunosuppressive conditions; (ii) presence of solid-organ transplantation (SOT); (iii) presence of hematological malignancies; (iv) the setting of PJP development (ICU vs. non-ICU). To reduce the selection bias of comparing patients with different pre-existing risk factors for PJP, the controls matching the cases’ age ± 10 years were selected with a simple randomization after such stratification.

### 2.2. Definitions

The diagnosis of PJP was considered according to the EORTC/MSGERC criteria: “proven” if *P. jirovecii* was detected with a direct immunofluorescence assay (DFA) on respiratory samples; “probable” in the presence of host factors, clinical features, and microbiological evidence; “possible” in the presence of host and clinical features with the absence of microbiological evidence [10]. According to the EORTC/MSGERC criteria, host factors were defined as: receipt of therapeutic doses of corticosteroids for at least 2 weeks within the past 60 days; antineoplastic, anti-inflammatory, or immunosuppressive treatment; low CD4 lymphocyte counts due to a medical condition, including, but not limited to, patients with primary immunodeficiencies, hematologic malignancies, SOTs, and allogeneic HSCT recipients [10].

Clinical features were considered as follows: any consistent radiographic features, particularly bilateral ground glass opacities; consolidations, small nodules, or unilateral infiltrates lobar in-filtrate; nodular infiltrates with or without cavitation; multifocal infiltrates; miliary pattern and/or respiratory symptoms with cough, dyspnea, and hypoxemia accompanying radiographic abnormalities, including consolidations, small nodules, unilateral infiltrates, pleural effusions, or cystic lesions on chest X-ray or computed tomography scan [10].

Microbiological evidence was defined as: detection of *Pneumocystis jirovecii* DNA by quantitative real-time polymerase chain reaction (PCR) in a respiratory tract specimen or beta-D-glucan ≥80 ng/L (pg/mL) detection in ≥2 consecutive serum samples provided that other etiologies have been excluded [10].

PJP severity was classified according to the 1996 classification by Miller at al., as mild, moderate, and severe, based on clinical features, peripheral oxygen saturation, chest radiology, and arterial oxygen tension (PaO2) in room air [13]. Cut-offs for arterial oxygen tension (PaO2) at rest and room air were: >11.0 kPa (>82.5 mmHg) for mild disease; 8.1–11.0 kPa (60.75–82.5 mmHg) for moderate disease; <8.0 kPa (<60 mmHg) for severe disease [13].

Charlson Comorbidity Index comorbidities were calculated to evaluate patients’ comorbidities [14]. Steroid dose was calculated as equivalent to dexamethasone, since dexamethasone was the most frequently used steroid drug in the studied population. Chronic steroidal treatment was defined as a daily dose ≥0.3 mg/kg of prednisone or equivalent for ≥2 weeks taken by the patient in the past 60 days, according to the EORTC/MSGERC criteria for host factors [10].

COVID-19 severity was assessed with the World Health Organization 9-point severity scale, as follows: 0—no clinical or virological evidence of infection; 1—ambulatory, no activity limitation; 2—ambulatory, activity limitation; 3—hospitalized, no oxygen therapy; 4—hospitalized, oxygen mask or nasal prongs; 5—hospitalized, noninvasive mechanical ventilation (NIMV) or high-flow nasal cannula (HFNC); 6—hospitalized, intubation and invasive mechanical ventilation (IMV); 7—hospitalized, IMV + additional support such as pressors or extracardiac membranous oxygenation (ECMO); 8—death [15].

### 2.3. Statistical Analysis

The statistical analysis was performed using SPSS version 27 (SPSS Inc., Chicago, IL, USA). Continuous variables were reported as the median and interquartile range and categorical variables as the frequencies and percentages. Categorial variables were confronted with the chi-squared test and Fisher’s exact test, when appropriate. Continuous variables were confronted with the Mann–Whitney U test. A significance level of 0.05 was set for the interpretation of the results. The confidence interval was set at 95% for the interpretation of the results. An univariate logistic regression analysis was performed to calculate the odds ratio (OR) between the development of PJP and the following demographic and clinical variables: age; sex; pregnancy; Charleson Comorbidity Index; SARS-CoV-2 vaccination; use of Pneumocystis prophylaxis; length of stay; COVID-19 disease duration; use of immunosuppressive treatment; type of immunosuppressive condition; length and dose of steroid and O2 therapy; worst recorded WHO grade and PaO2/FiO2 ratio; lowest absolute lymphocyte value and highest LDH, CRP, and D-dimer value. A forward stepwise logistic regression was used to identify the independent risk factors of PJP out of the variables resulted significant from the univariate analysis. At each step, variables were added according to their contribution to the model’s R^2^, and the *p*-value threshold of 0.05 was set to the limit on the total number of variables included in the final model.

## 3. Results

We enrolled 54 patients (18 cases and 36 matched controls) from a total of 380 patients admitted during the study period (14%). The median age of the included patients was 60 years (95% IQR 51–68), with 34.5% of females and 12.7% of pregnant women, and a median Charlson index of 3 (95% IQR 1–5). The largest part of the patients (60%) had received at least one dose of the SARS-CoV-2 vaccination and stayed in the hospital for a median of 16 days (95% IQR 12–34). During the hospital stay, the patients received a median of 85 mg (95% IQR 55–180) cumulative dexamethasone dose for a median of 15 days (95% IQR 12–27), and only 6 out of 54 (11%) received immunomodulatory therapies for COVID-19, in detail: 4 received baricitinib and 2 received tocilizumab. Half of the studied population (53%) was immunocompromised for hematologic malignancies, with the most common condition represented by non-Hodgkin lymphoma (Table 1).

Among 18 cases of PJP, 16 were diagnosed as “proven”, of which 10 were diagnosed with DFA on BAL, while 6 were diagnosed with DFA from nonbronchoscopic-obtained lower respiratory tract samples (bronchial aspiration or mini-BAL). Regarding the two patients diagnosed as having “possible” PJP, they presented both host and radiological factors, and no minor microbiological criteria (PCR from respiratory samples or serum BDG). Seven of the eighteen cases were immunocompromised, of which five (27.8%) suffered from hematologic conditions and two (11%) were SOT recipients, while the other patients had no previous immunological impairment. Nine out of eighteen cases of PJP required ICU admission; among them, four were diagnosed before ICU admission and five during the ICU stay.

All patients but one were diagnosed with a moderate-to-severe course of PJP according to the Miller criteria, and were treated with intravenous trimethoprim–sulfamethoxazole (TMP-SMX) at a dose of 15–20 mg/kg, divided into three to four daily doses for 21 days (with a switch to oral drug whenever it was feasible after reaching clinical improvement), plus prednisone at a dose of 40 mg (or dexamethasone at equivalent dose) twice daily for the first 5 days, followed by 40 mg daily for 5 days and 20 mg daily for the remaining 11 days. One patient presented with mild disease and was treated with oral TMP-SMX 2 double strength tabs three times daily for 21 days with no adjunctive steroids. Only one patient diagnosed with PJP had detectable serum BDG in two samples obtained on different days (>523 pg/mL in both determinations). The patient had follicular lymphoma treated with rituximab, a severe course of PJP requiring ICU admission, and died 22 days after PJP diagnosis; he did not develop any concurrent IFIs during the hospital stay, to our knowledge. Conversely, two patients developed concurrent IFIs, namely one case of *Aspergillus* tracheobronchitis and one case of invasive pulmonary aspergillosis. They were both pregnant patients in their early thirties, both requiring ICU admission, and both died. No patients developed drug adverse reactions that required the cessation of the therapy with TMP-SMX, and six of them (33%) died.

Patients diagnosed with PJP had a similar comorbidity index, WHO grade, and number of days of the highest O2 support received than the controls; conversely, the length of stay, mortality, and ICU admissions were higher in cases compared to the controls, although nonsignificantly. Also, the nadir of the lymphocyte count and of the PaO2/FiO2 ratio recorded were lower in PJP patients compared to the controls (Table 1).

Compared to the controls, patients with PJP had significantly lower median lymphocyte values (540 vs. 780 cells/mm^3^, *p* = 0.033), longer COVID-19 disease duration (25 vs. 16 days, *p* = 0.014), a higher cumulative dose of steroid received (178.5 vs. 78 mg, *p* = 0.026), higher CRP values (14.4 vs. 6.3 mg/dL, *p* = 0.005), and a lower SARS-CoV-2 vaccination rate than the controls (7 patients with at least one dose vs. 26 patients with no history of vaccination, *p* = 0.029) (Table 1).

From the univariate analysis, the cumulative steroid dose received during the hospital stay (OR 1.004, 95%CI 1–1.008, *p* = 0.042) and the highest CRP value recorded during the hospitalization (OR 1.076, 95%CI 1.016–1.140, *p* = 0.012) were identified as risk factors for PJP, while SARS-CoV-2 vaccination with one (OR 0.269, 95%CI 0.083–0.877, *p* = 0.029) and two doses (OR 0.304, 95%CI 0.093–0.994, *p* = 0.049) was identified as a protective factor for PJP. In order to identify factors that may predict the development of PJP, a forward multivariate logistic regression analysis was conducted by simultaneously entering the cumulative steroid dose received, the highest CRP value, and vaccination with at least one dose into the model. The results indicated that the model accounted for a significant amount of variance in success (likelihood ratio: chi-squared = 5.24, *p* = 0.022). However, out of all the predictors in the model, only the cumulative steroidal dose resulted as an independent predictor of PJP development (b = 0.04, SE = 0.02, OR = 1.004, 95%CI = 1–1.008, *p* = 0.042) after controlling for SARS-CoV-2 vaccination and the CRP value.

By considering the clinical differences between vaccinated (at least one dose) and unvaccinated patients, unvaccinated patients had a significantly lower recorded P/F ratio (*p* = 0.015) and received significantly higher doses of steroids (*p* = 0.031) than vaccinated patients (Table 2).

## 4. Discussion

Compared to the controls, patients with PJP have significantly lower median lymphocyte values, longer COVID-19 disease duration, a higher cumulative dose of steroids received, higher CRP values, and a lower SARS-CoV-2 vaccination rate, although only the cumulative steroid dose received resulted to be independently associated with the risk of developing PJP.

As we know from years of study about PJP, the main risk factors for its development are: HIV infection with a CD4+ lymphocyte count <200 cells/mm^3^ and high HIV viral loads; certain hematologic conditions (graft versus host diseases, the use of prolonged lymphopenia in the context of other hematologic malignancies, the use of lymphocyte-depleting agents for lymphomas or lymphoblastic leukemia, etc.); autoimmune conditions treated with lymphocyte-depleting agents; high and prolonged doses of steroids (≥0.3 mg/kg prednisone equivalent for ≥2 weeks in the past 60 days); solid-organ transplantation, especially if the lung or small bowel, or complicated by lymphopenia or the use of antithymocyte agents [16,17,18].

On the other hand, it is well described how the occurrence of a severe acute respiratory illness (SARI), especially when it ends up in acute respiratory distress syndrome (ARDS) requiring ICU admission, becomes an ideal breeding ground for PJP development due to lung damage, sepsis-induced immune-paralysis, and iatrogenic immunosuppression in otherwise immunocompetent hosts, as reported by Beumer and colleagues during the influenza A epidemic of 2015, when 2% of non-ICU and 7% of ICU patients developed PJP as a complication of viral infection [19].

Similar mechanisms have been called upon concerning the development of IFIs in COVID-19 patients, together with the use of corticosteroids and immunomodulatory agents as a part of therapy and with the absolute count and CD4+ lymphopenia associated with COVID-19, whose levels correlate with disease severity and poor prognosis [20,21,22].

According to these considerations, patients with COVID-19, even if previously immunocompetent, may develop de novo risk factors for PJP, such as severe lymphopenia, the use of high doses of steroids, or other immunomodulating drugs, together with COVID-19-associated pathological mechanisms (i.e., lung epithelial barrier damage and immune system dysregulation) [23].

In fact, considering the results of the systematic reviews by Amstutz and Sasani, only half of the patients from the included studies were found to have at least one immunosuppressive condition before COVID-19 onset that predisposed them to PJP, most commonly HIV infection, while the remaining patients had no classic risk factors for PJP development, meaning that COVID-19 itself and its treatment are a novel risk factor for PJP to be addressed [11,12].

Moreover, Amstutz and colleagues have found that 45% of patients with PJP received long-term corticosteroids before and during COVID-19, and discovered a median absolute lymphocyte count (IQR) of 610 (280–920) cells/mm^3^ (*n* = 23) and a CD4 count (IQR) of 66 (33–291.5) cells/mm^3^ (*n* = 20) in the studied population [11].

In light of the abovementioned considerations, the results of our study enforce the hypothesis that immunocompetent patients can also acquire de novo risk factors for PJP. In fact, in these patients, the amount of received steroids is the main risk factor for PJP development. It is noteworthy that the other known risk factors of PJP resulted to be nonsignificant in our study (i.e., severity of COVID-19, lymphopenia, chronic immunosuppression, and ICU stay). Notably, in the cohort of patients of our study that developed PJP, the median length of steroid administration was 15 (IQR 12–20) days, i.e., within the range of two weeks established by the EORT/MSGERC for host factors [10].

Moreover, it must be noted that, according to previous data from the literature, there is a wide variability among healthy subjects in lymphocyte response to corticosteroid (−6.7% to 99.7%), and also that an intermittent use of a high dose of corticosteroids, not followed by daily administration, as happens in chemotherapy regimens for lymphoma, is associated with the risk of developing PJP [24,25]. Thus, in COVID-19 patients, the intermittent use of a high dose of glucocorticoids can also become an ideal breeding ground for PJP, as demonstrated by our results.

The underlying mechanism behind the role of corticosteroids in increasing the risk of PJP has been explained by some experiments on rats carried in the late 1990s. In fact, the administration of corticosteroids has been used for decades for inducing PJP in experimental rats [26]. In rats with corticosteroid-induced PJP, researchers have observed a depletion of T helper cells in the blood, as well as a fall in T helper (Th) cells and a rise in T suppressor (Ts) cells in the lungs, with a normal subpopulation in the thymus, spleen, and bone marrow [27]. When corticosteroids were withdrawn, the rats gained weight, cleared *P. jirovecii* from the lungs, and regenerated their lymphoid tissues; the lymphocyte subpopulations exhibited variation in their frequencies at different body sites, but gradually returned to baseline levels [27]. It would be interesting to know with ad hoc studies if the length or the cumulative steroid dose can also have an impact on the COVID-19-associated PJP outcome, as shown, for example, in the ANSWER cohort study by Shiba and colleagues in patients with rheumatoid arthritis that developed PJP, in which a baseline higher dose of chronic steroid at the time of PJP diagnosis was associated with higher mortality [28].

It must be highlighted that most patients in our cohort developed PJP during the first waves of the pandemic (at the beginning of the vaccination campaign), when most of them had not completed the vaccination cycle; it was a time in which general practitioners in Italy used to prescribe prolonged corticosteroid therapy at the onset of COVID-19 symptoms, irrespective of the need of oxygen supplementation. This could be the reason why, in our cohort, the lack of vaccination and higher CRP nonindependently resulted to be associated with PJP, together with steroid use: unvaccinated patients had more severe disease than vaccinated patients, as we demonstrated in Table 2, and, probably because of that, reached a higher CRP value than vaccinated individuals, and thus received higher steroid doses.

Despite the findings of our study, we speculate that other elements, independent from the universally accepted risk factors, must also be considered to explain the development of PJP in COVD-19 patients, especially given the fact that some clinical centers have never diagnosed a single case of PJP in patients diagnosed with COVID-19.

One of these elements could be the presence of the prior colonization of the respiratory tract by *Pneumocystis jirovecii* spores before hospitalization, or the airborne transmission from infected patients to susceptible ones, as occurred in the past with PJP outbreaks in transplantation hospital units [29,30]. The clustering transmission of *Pneumocystis jirovecii* in the hospital setting can explain why the reports of PJP cases are so heterogeneous in the literature, with some centers such as ours reporting higher numbers of cases than others, but such a hypothesis must be confirmed with ad hoc studies.

The PCR for *P. jirovecii* on respiratory samples, noninvasively obtained, such as with sputum, tracheal aspirate, or oral washing, also have a promising role in identifying colonized patients at risk of developing PJP, even though we think that the DFA should be preferred over the PCR for the diagnosis of PJP, given the low efficacy of the PCR in discriminating between colonization and infection [31].

Surprisingly, despite all but two patients in our cohort receiving a “proven” diagnosis of PJP, and only two receiving a “possible” diagnosis, no patients except for one had a detectable BDG on repeated serum samples, in line with the findings of Amstutz and colleagues, which reported a positive BDG only in 66.7% of patients studied in their meta-analysis [11]. These findings are apparently in contrast to the high negative predictive value of the BDG to rule out the diagnosis of PJP, as reported in several studies [32]. Nonetheless, according to the meta-analysis published in 2020 by Del Corpo and colleagues, the BDG test is more sensitive in patients with HIV than in those without (94% versus 86%), and a negative BDG is only associated with a low post-test probability of PJP (≤5%) when the pretest probability was low to intermediate (≤20% in non-HIV and ≤50% in HIV) [33]. The low rate of BDG positivity in COVID-19 patients can be related to a lower fungal burden in such populations compared to HIV patients, but this hypothesis needs to be confirmed with specific studies.

As for indications for PJP primary chemoprophylaxis, no definite rules exist for non-HIV patients, and no studies have ever been conducted in patients with COVID-19. The 5th European Conference on Infections in Leukaemia (ECIL-5) established that the main indications of PJP prophylaxis are acute lymphoid leukemia, allogeneic HSCT, treatment with alemtuzumab, fludarabine/cyclophosphamide/rituximab combinations, >4 weeks of treatment with corticosteroids, and well-defined primary immune deficiencies in children [17]. On the other hand, the European League Against Rheumatism (EULAR) recommends PJP prophylaxis in patients treated with daily doses of >15–30 mg of prednisolone or an equivalent for >2–4 weeks [34].

Surprisingly, in our study, neither the PJP prophylaxis result as a protective factor for PJP development (OR 0.252, 95%CI = 0.029–2.226, *p* = 0.215) nor chronic steroid therapy, SOT, and malignancies resulted as risk factors. Thus, it is very difficult to identify COVID-19 patients at risk of developing PJP early that may benefit from chemoprophylaxis. Generalizing our results, despite that we could not find any particular subgroup of COVID-19 patients at high risk of PJP development, we can say that PJP can occur in any COVID-19 patient who received corticosteroids for at least 2 weeks, regardless of the previous immune status. Thus, it is important to suspect PJP in patients who present with a persistent or worsening of relapse of SARI, even if they achieved virological clearance from SARS-CoV-2.

It is also important to identify on admission those patients for which PJP prophylaxis is indicated independently from COVID-19, and prescribe chemoprophylaxis during the hospital stay and after discharge, if needed, especially if they are expected to receive corticosteroids for COVID-19 for a period longer than 2 weeks.

The present study has several limitations: the retrospective design; the small number of subjects involved, given the rarity of the disease; data collection from a single center in a limited period of time; potential selection bias.

## 5. Conclusions

PJP develops in COVID-19 patients regardless of previous immunosuppressive conditions. High corticosteroid doses received before and during hospital admission is the independently associated risk factor for PJP development in this population. The BDG assay seems to have a poor performance in ruling out PJP in COVID-19 patients; therefore, bronchoscopy must not be withheld in cases of high suspicion of pulmonary superinfection or coinfection, especially in patients who received high doses of corticosteroids.

Further studies are needed to establish the incidence and risk factors of *Pneumocystis jirovecii* colonization among patients hospitalized for COVID-19, and the risks for the development of symptomatic infection among colonized hosts, also aiming to identify high-risk patients that might benefit from PJP chemoprophylaxis.

## Figures and Tables

**Table 1 jof-09-00838-t001:** Overall demographic and clinical characteristics of the population, group comparison, and univariate and multivariate analysis. ICU: intensive care unit; CRP: C-reactive protein.

	Overall N = 54	Cases N = 18	Controls N = 36	*p*-Value	OR (95%CI)	*p*-Value
Age, years, median (IQR)	60 (51–68)	60 (49.75–70)	60 (49–78)	0.98	0.994 (0.995–1.034)	0.754
Females, *n* (%)	19 (34.5)	7 (39)	12 (32)	0.764	0.754 (0.234–2.433)	0.637
Pregnant, *n* (%)	7 (12.7)	4 (22)	3 (8)	0.2	3.2 (064–16.32)	0.155
Charlson Comorbidity Index, median (IQR)	3 (1–5)	3 (0.75–5)	3 (1–5)	0.389	0.876 (0.685–1.12)	0.291
Deaths, *n* (%)	14 (25.5)	6 (33)	8 (21.6)	0.51	-	-
ICU admission, *n* (%)	20 (37)	9 (50)	11(30)	0.232	-	-
SARS-CoV-2 vaccination, *n* (%)						
1 dose	33 (60)	7 (39)	26 (70)	0.027	0.269 (0.083–0.877)	0.029
2 doses	29 (52)	6 (33)	23 (62)	0.042	0.304 (0.093–0.994)	0.049
3 doses	16 (29)	3 (16.7)	13 (35)	0.135	0.369 (0.9–1.5)	0.166
4 doses	2 (3.6)	0 (0)	2 (5.4)	1	0 (0–0)	-
*Pneumocystis jirovecii* prophylaxis recipients, *n* (%)	8 (14.5)	1 (5.6)	7 (19)	0.25	0.252 (0.029–2.226)	0.215
Length of stay, days, median (IQR)	16 (12–34)	20.5 (14.5–50.25)	15 (10.5–27)	0.058	1.033 (0.998–1.068)	0.062
Days of SARS-CoV-2 positivity, median (IQR)	20 (14–26)	25 (20–37)	16 (13–23)	0.014	1.141 (0.993–1.322)	0.062
Use of immunomodulatory drug for COVID-19, *n* (%)	6 (11)	3 (16.7)	3 (8)	0.381	2.267 (0.0409–12.5)	0.349
Hematologic malignancy, *n* (%)	29 (53)	5 (27.8)	10 (27)	0.39	0.564 (0.166–1.915)	0.359
Use of anti-CD20, *n* (%)	6 (11)	3 (16.7)	3 (8)	0.381	2.267 (0.409–12.5)	0.349
Chronic steroidal treatment, *n* (%)	8 (14.5)	3 (16.7)	5 (13.5)	1	1.28 (0.270–6)	0.756
Solid organ transplant recipients, *n* (%)	6 (11)	2 (11)	4 (10.8)	1	1.031 (0.171–6.23)	0.973
Cumulative steroid dose during hospital stay, milligrams, median (IQR)	84 (55–190)	178.5 (68–513)	78 (46–158)	0.026	1.004 (1–1.008)	0.042
Number of days of steroid, median (IQR)	15 (12–27)	15 (12–20)	16.5 (12–29.25)	0.718	0.984 (0.929–1.043)	0.596
Worst WHO grade, median (IQR)	4 (4–6)	5 (4–6)	4 (4–6.5)	0.434	1.287 (0.7–2.378)	0.421
Lowest PaO2/FiO2 ratio, median (IQR)	135 (89–235)	100 (65–191)	150 (100–269)	0.081	0.995 (0.988–1.002)	0.13
Days of highest O2, *n* (%)	7 (5–10)	7 (5–10)	6 (4.75–10)	0.849	0.965 (0.873–1.068)	0.5
Lowest lymphocyte value, cells/mm^3^, median (IQR)	620 (320–1480)	540 (217–772)	780 (415–2313)	0.033	1 (0.999–1)	0.732
Highest CRP value, mg/dL, median (IQR)	10 (2.9–20)	14.4 (10–28.6)	6.3 (2.3–15)	0.005	1.076 (1.016–1.140)	0.012
Highest LDH value, IU/mL, median (IQR)	384 (289–524)	368 (289–452)	384 (258–533)	0.788	0.999 (0.996–1.002)	0.4
Ferritin on admission, median (IQR)	595 (246–1335)	740 (279–1342)	520 (205–1073)	0.332	1 (0.999–1.001)	0.588
Highest D-dimer value, median (IQR)	765 (533–1524)	761 (497–1392)	1051 (570–1526)	0.554	1 (1)	0.641

**Table 2 jof-09-00838-t002:** Differences in vaccinated and nonvaccinated patients in variables associated with COVID-19 severity.

	Vaccinated N = 32	Non-Vaccinated N = 22	*p*-Value
Cumulative steroid dose during admission, milligrams, median (IQR)	78 (50–163)	168.5 (80–568)	0.031
Highest CRP value, mg/dL, median (IQR)	8.26 (2.34–21.59)	12.36 (3.33–17.37)	0.223
Worst WHO grade, median (IQR)	5 (5–6.5)	6 (5–7)	0.089
Lowest PaO2/FiO2 ratio, median (IQR)	183 (100–275)	100 (75–140)	0.015
Days of highest O2, *n* (%)	6.5 (4.25–9.75)	7 (5–10)	0.739
Highest LDH value, IU/mL, median (IQR)	348 (221–528)	390 (305–505)	0.460
Lowest lymphocyte value, cells/mm^3^, median (IQR)	600 (315–1635)	680 (355–1065)	0.945

## Data Availability

The data that support the findings of this study are available from the Federico II University Hospital, but restrictions apply to the availability of these data, which were used under license for the current study, and so are not publicly available. Data are however available from the authors upon reasonable request and with permission of the Federico II University Hospital.

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
