# Peer review of "Risk Factors for Pneumocystis jirovecii Pneumonia in Non-HIV Patients Hospitalized for COVID-19: A Case-Control Study"

_jof, 2023, doi:10.3390/jof9080838_

Round 1

Reviewer 1 Report (Previous Reviewer 1)

The manuscript is now improved. Some problems in the design of the references should be fixed like: Line 225, 249, 259 etc

Some improvements are required

Author Response

Thank you. The references' design has been fixed.

Reviewer 2 Report (New Reviewer)

The manuscript investigates the risk factors for Pneumocystis jirovecii pneumonia (PCP) in non-HIV patients hospitalized for COVID-19. The authors conducted a retrospective case-control study in a cohort of patients with confirmed COVID-19, identifying risk factors associated with PCP development. The results highlight that PCP can occur in COVID-19 patients regardless of pre-existing immunosuppressive conditions and disease severity. The study found that the cumulative dose of corticosteroids received during hospitalization is the independent risk factor for PCP development.

1. Abstract: The abstract provides a comprehensive overview of the study, but some parts could be made more concise and reader-friendly without compromising essential information. Please, include odds ratio to understand the association degree between corticosteroids and PCP development.

2. Background Section: The background section provides a comprehensive overview of COVID-19 and its associated infections, but it could benefit from a more focused discussion on the relevance of PCP in the context of COVID-19. You may want to emphasize the rarity of PCP in COVID-19 patients and the need for more research in this area. Additionally, consider adding a concise summary of previous studies related to PCP in COVID-19. How relevant are other IFIs in COVID-19? Suggested literature: 10.3201/eid2704.204895, 10.1007/s11046-023-00770-w, 10.1093/mmy/myad072

3. Methodology: Provide more details on patient selection criteria and the rationale for choosing specific control groups. Clarify how patients with confirmed PCP were identified and the methods used for matching controls. Additionally, mention any potential sources of bias and how they were addressed.

4. Statistical Analysis: The manuscript would benefit from a more in-depth discussion of the statistical methods used, particularly in the univariate and multivariate logistic regression analyses.

5. Results: While the results are well-presented, it would be beneficial to provide confidence intervals for the odds ratios in the univariate logistic regression analysis. This would add to the statistical interpretation of the findings.

6. Discussion: The discussion section could further elaborate on the potential mechanisms underlying the association between corticosteroid use and PCP in COVID-19 patients. Address whether the steroid-induced immunosuppression is the primary driver or if other factors contribute to PCP development.

7. Limitations: It would be beneficial to acknowledge and discuss the study's limitations, such as the retrospective nature and potential selection bias.

8. Generalizability: Discuss the generalizability of the findings to other populations and settings. Are there any specific patient subgroups that might be more susceptible to PCP in the context of COVID-19?

9. Recommendations and Future Research: End the manuscript with a section discussing the clinical implications of the findings and potential areas for future research. Suggest strategies to minimize the risk of PCP in COVID-19 patients while using corticosteroids for treatment.

10. Ethical Considerations: It is essential to include information about ethical approval and patient consent, especially in retrospective studies involving human subjects.

just minor errors

Author Response

  1. We appreciate the comment and we improved the abstract as suggested
  2. We improved background section according to reviewer’s suggestions.
  3. We appreciate the suggestion and we improved the methods accordingly
  4. We thanks the reviewer for this suggestion and we further discussed the statistical analysis we performed
  5. We appreciate the comment, we displayed all the confidence intervals for the OR both in the Table and in the text. We added CI in the abstract as well during this revision.
  6. We thank the reviewer for this suggestion and we accordingly discuss the pathogenetic role of corticosteroids in the development of pneumocystosis (lines 289-298).
  7. According to the recommendations, we highlighted the study limitations at the end of the discussion section
  8. We appreciate this suggestion and we further discuss generalizability of our results in lines 355-364
  9. Please find this issue better discussed at the end of discussion section and in the conclusions.
  10. Please find the ethical statement before the reference section. We included a statement about the nature of data and the consent for publication.

Reviewer 3 Report (New Reviewer)

This study investigated the risk factor of developing PJP among hospitalized patients with COVID-19. This study is interesting and the manuscript is well-written. Thus, I just have several minor suggestions.

1. Please clarify the fullname of COVID-19 and replaced PCP by PJP

2. What do you mean by yellowish marker

3. Please analyzed the confounding effect of antiviral agents, such as remdesivir, and the IL-6 blockade.

Author Response

  1. We thank the reviewer for the comment. Please find COVID-19 full name in the first line of background section. PCP was changed in PJP in all the document.
  2. Yellow marks were included to underlying some changes during the first review process and have been deleted all over the document.
  3. We appreciate the reviewer’s comment. Unfortunately, only 2 patients in our cohort (both in control group) received IL-6 blocking agents, the rest baricitinib, so the role of IL-6 as confounding agents cannot be assessed. We assessed the role of immunomodulatory agents in general in Table 1 and we found no significant differences between cases and controls in the risk of PJP development. We did not collect data on antiviral therapy for COVID in our population since we didn’t consider a priori antivirals as a potential variable that can explain differences in PJP development.

This manuscript is a resubmission of an earlier submission. The following is a list of the peer review reports and author responses from that submission.

Round 1

Reviewer 1 Report

1. The background (introduction) should be summarized and the authors need to enhance its flow, it is currently closer to a discussion section.

2. The discussion is poorly written and hard to read.

3. The importance of real time PCR is underestimated.

4. The authors performed minor literature searching, below are some recent articles and recent comprehensive review:

Alsayed, A. R., Al-Dulaimi, A., Alkhatib, M., Al Maqbali, M., Al-Najjar, M. A. A., & Al-Rshaidat, M. M. (2022). A comprehensive clinical guide for Pneumocystis jirovecii pneumonia: a missing therapeutic target in HIV-uninfected patients. Expert Review of Respiratory Medicine, 1-24.  

Alsayed, A. R., Talib, W., Al-Dulaimi, A., Daoud, S., & Al Maqbali, M. (2022). The first detection of Pneumocystis jirovecii in asthmatic patients post-COVID-19 in Jordan. Bosnian Journal of Basic Medical Sciences, 22(5), 784.  

Amstutz, P., Bahr, N. C., Snyder, K., & Shoemaker, D. M. (2023, February). Pneumocystis jirovecii infections among COVID-19 patients: a case series and literature review. In Open Forum Infectious Diseases (Vol. 10, No. 2, p. ofad043). US: Oxford University Press.   Shiba, H., Kotani, T., Nagai, K., Hata, K., Yamamoto, W., Yoshikawa, A., ... & Takeuchi, T. (2023). Prognostic Factors Affecting Death in Patients with Rheumatoid Arthritis Complicated by Pneumocystis jirovecii Pneumonia and One-Year Clinical Course: The ANSWER Cohort Study. International Journal of Molecular Sciences, 24(8), 7399.

Should be improved

Reviewer 2 Report

Risk factors for Pneumocystis jirovecii pneumonia in non-HIV patients hospitalized for COVID-19: a case-control study

Comments: The authors reported an important issue of PCP in COVID-19 patients. However, the logic of analysis methods was not clearly described and the final model seemed not to reach any significant factor associated with PCP.

1.      Line 109: diagnosed with PCP during the admission. (hospital stay?)

2.      In the Method section of the text, you describe the detection of Pneumocystis jirovecii DNA by quantitative real-time polymerase chain reaction (PCR) as diagnosis criteria but do not describe using direct immunofluorescence assay on respiratory samples, which was mentioned in the abstract to diagnose PCP.

3.      It seems that ICU stay was used as ICU admission throughout the text, please revise it, if any.

4.      Line 216, the highest CRP value recorded during the admission (hospitalization?)

5.      The methods of the statistical analysis section were grossly inadequate. The statistical methods such as how the logistic regression models were performed between group analyses and why to do the secondary analyses reported in the results section have not been described in the methods section. The authors should briefly describe the types and reasons for analyses performed. In your analysis section, can you let your reader know the precise approach (forward, backward, or stepwise) in the multivariable logistic regression modeling you applied and state exactly what the priori confounders were?

6.      Table 1 is of less value and could be put into Table 2 to compare the difference between cases and controls. Then you may separate the OR relevant data into Table 2 with OR, 95%CI, and p in different columns. A second aOR might be tried in the absence of potential confounders of vaccination factors.

7.      Line 229-233, Although Chi-square difference analysis and univariate analysis showed some significant factors, the final model of multivariable analysis did not find any significant factors associated with PCP, possibly as confounded by the low vaccination factor. The authors should clearly describe or try the forward approach in the multivariable regression model to remove potential confounders in the final model analysis.

8.      Discussion was too redundant with many sentences already described in the introduction section. The authors should mainly focus on discussing the results obtained in the study.